# Vehicle Detection and Tracking with Roadside LiDAR Using Improved ResNet18 and the Hungarian Algorithm

**DOI:** 10.3390/s23198143

**Published:** 2023-09-28

**Authors:** Ciyun Lin, Ganghao Sun, Dayong Wu, Chen Xie

**Affiliations:** 1Department of Traffic Information and Control Engineering, Jilin University No. 5988, Renmin Street, Changchun 130022, China; linciyun@jlu.edu.cn (C.L.); sungh20@mails.jlu.edu.cn (G.S.); xiechen19@mails.jlu.edu.cn (C.X.); 2Jilin Engineering Research Center for Intelligent Transportation System, Changchun 130022, China; 3Texas A&M Transportation Institute, 12700 Park Central Dr., Suite 1000, Dallas, TX 75251, USA

**Keywords:** autonomous driving, vehicle detection, vehicle tracking, ResNet18, roadside LiDAR sensor, KITTI, MATLAB/Simulink

## Abstract

By the end of the 2020s, full autonomy in autonomous driving may become commercially viable in certain regions. However, achieving Level 5 autonomy requires crucial collaborations between vehicles and infrastructure, necessitating high-speed data processing and low-latency capabilities. This paper introduces a vehicle tracking algorithm based on roadside LiDAR (light detection and ranging) infrastructure to reduce the latency to 100 ms without compromising the detection accuracy. We first develop a vehicle detection architecture based on ResNet18 that can more effectively detect vehicles at a full frame rate by improving the BEV mapping and the loss function of the optimizer. Then, we propose a new three-stage vehicle tracking algorithm. This algorithm enhances the Hungarian algorithm to better match objects detected in consecutive frames, while time–space logicality and trajectory similarity are proposed to address the short-term occlusion problem. Finally, the system is tested on static scenes in the KITTI dataset and the MATLAB/Simulink simulation dataset. The results show that the proposed framework outperforms other methods, with F1-scores of 96.97% and 98.58% for vehicle detection for the KITTI and MATLAB/Simulink datasets, respectively. For vehicle tracking, the MOTA are 88.12% and 90.56%, and the ID-F1 are 95.16% and 96.43%, which are better optimized than the traditional Hungarian algorithm. In particular, it has a significant improvement in calculation speed, which is important for real-time transportation applications.

## 1. Introduction

In the coming decades, autonomous driving will become sufficiently reliable, affordable, and pervasive to ultimately displace most human driving. Optimists predict that full autonomy may be commercially available and legal to use in some jurisdictions in the late 2020s [1]. However, autonomous driving still faces great challenges to full autonomy due to the limitations in global perspectives, occlusions, and long-range perception capacity if only relying on autonomous vehicle (AV)’s in-vehicle sensors. It has been widely agreed that vehicle–infrastructure cooperation is required to achieve Level 5 autonomy [2]. The infrastructure leveraged by roadside sensors can assist an autonomous vehicle with driving by offloading perception and planning from the vehicle to roadside infrastructure. To achieve this goal, the infrastructure system must be able to process voluminous roadside sensor data to obtain enriched traffic information at a full frame rate, i.e., with a tail latency of less than 100 milliseconds, without sacrificing accuracy [3].

To assist AVs’ driving in a complex multimodal traffic environment, it is essential to obtain real-time, high-accuracy, high-resolution trajectory data for all road users from the infrastructure. However, current intelligent transportation system (ITS) sensors such as loop detectors, video cameras, microwave radars, and Bluetooth sensors mainly provide macro-traffic data, namely traffic flow rates, average speeds, and occupancy [4]. As a type of active vision sensor, light detection and ranging (LiDAR) has advantages over other types of sensors due to its insensitivity to external light changes, strong adaptability to complex environments, high accuracy and frequency, wide coverage, and enriched depth information [5]. LiDAR has great potential, especially in extracting high-resolution micro traffic data through 360-degree detection of the surrounding environment in real-time [6]. When deployed at the infrastructure, LiDAR can obtain high-resolution, fine-grained positions, directions, speeds, and even the trajectory of each road user within the scanning range, which could be used as a valuable data input for vehicle-to-everything (V2X) and cooperative vehicle infrastructure systems (CVISs) [7].

Currently, the most typical application of LiDAR is for detecting road and traffic information when installed in AVs. However, with ongoing advancements in vehicle-to-infrastructure (V2I) technology, there has been a surge in research focusing on the use of roadside LiDAR for vehicle detection and tracking [8,9,10]. Despite the great promise of these studies, most remain at the research stage, encountering challenges in real-time applications. To address this, we leverage deep learning-based object detection algorithms used in AVs, improving and optimizing them to enhance the real-time detection capabilities of roadside LiDAR.

In this paper, we describe the algorithms and optimizations we have developed that enable infrastructure with roadside LiDAR to detect and track vehicles in real-time. We first develop a vehicle detection architecture based on ResNet18 that can more effectively detect vehicles at a full frame rate by improving the BEV (bird’s eye view) mapping and the loss function of the optimizer. Then, we propose a new three-stage vehicle tracking algorithm. This algorithm enhances the Hungarian algorithm to better match objects detected in consecutive frames, while time–space logicality and trajectory similarity are proposed to address the short-term occlusion problem. Finally, the system is tested on static scenes in the KITTI dataset [11] and the MATLAB/Simulink simulation dataset. The results show that the proposed framework outperforms other methods in detection and tracking and significantly improves the calculation speed for real-time applications. The main innovations of our work are described as follows:

(1) A vehicle detection architecture based on ResNet18 is proposed, which considers the vehicle’s travel area during BEV mapping and mitigates overhead interferences. The improved loss function accounts for both the overlapping volume and similarity between predicted and actual bounding boxes, ensuring an improved detection accuracy.

(2) A three-stage vehicle tracking algorithm is proposed to track the trajectory of the detected vehicles.

(3) An improved Hungarian algorithm is developed by adding the IoU metric as the efficiency matrix for better matching of objects detected in consecutive frames.

(4) Time–space logicality and trajectory similarity strategies are proposed to solve the short-time occlusion problem. These strategies are designed to maintain tracking accuracy even in scenarios where the tracked vehicle is temporarily occluded, ensuring the robustness of our system.

After the introduction, the rest of this paper is organized as follows: Section 2 summarizes related work and Section 3 describes the proposed method of vehicle detection and tracking. In Section 4, the case study is presented, in which experiments are conducted to illustrate the performance of our method. The results are also analyzed and compared with other models in this section. This study is concluded in Section 5, with remarks on future research.

## 2. Related Work

### 2.1. Vehicle Detection and Tracking Using In-Vehicle LiDAR

Vehicle detection and tracking are essential for automated driving. Through accurate vehicle detection and tracking, the safety of self-driving vehicles can be ensured, and traffic accidents can be avoided. By monitoring the position, speed, and trajectory of surrounding vehicles in real-time, the automatic driving system can make intelligent decisions [12] and plan optimal paths [13]. In recent years, LiDAR has been gradually used as a perception sensor for automatic driving due to its advantages such as anti-interference, light-sensitive invariance, and a high accuracy [14].

The vehicle detection method based on in-vehicle LiDAR is mainly a deep learning method which uses the point cloud data as direct inputs to generate a 3D bounding box for each detected object [15]. Depending on the data representation and modeling basics, models can be classified into three main categories: point-based methods (e.g., PointNet [16] and PointNet++ [17]), voxel-based methods (e.g., VoxelNet [18]), and BEV-based methods (i.e., PIXOR [19]). PointNet is the pioneering, work with raw point clouds used as the input for deep learning. It provides a unified architecture for applications ranging from object classification to part segmentation and scene semantic parsing [16]. Since PointNet has limited performance in complicated scenes, PointNet++ is proposed to capture the local features of neighboring points and improve the original network’s performance [17]. VoxelNet implements voxelization in 3D space and extracts features in grouped voxels by using 3D convolutions. It is widely used for 3D object detection by using a generic 3D detection network that unifies feature extraction and bounding box prediction into a single-stage, end-to-end trainable deep network [18]. In a comparative study, the conclusion that PointNet is useful and has high performance in small scenarios, while VoxleNet is more useful in large scenarios, when tested using the KITTI dataset [20]. PIXOR, as a BEV-based method, is a proposal-free, single-stage detector that outputs oriented 3D object estimates decoded from pixel-wise neural network predictions. It is more efficient to use the 3D data by representing the scene from the BEV and reducing the computation cost due to the high dimensionality of point clouds [18]. BEV-based methods can run at high speeds, but with reduced accuracy. They also lose the Z-value of 3D measurements when the point cloud is projected to the X-o-Y plane.

For vehicle tracking, the distance characteristics of objects between adjacent frames are commonly used in mainstream methods, such as the Hungarian method [21]. Predictive tracking algorithms using Kalman and particle filtering are also another major type of method [22,23,24].

However, it is difficult to obtain a global view just by relying on the sensors equipped on the AV. And due to the occlusion of the vehicle in front, it will greatly limit the remote sensing ability of autonomous driving. To deal with these challenges, roadside sensing has become an important solution. Through V2I technology, roadside sensing can provide more comprehensive information and data for AVs [25].

### 2.2. Vehicle Detection and Tracking with Roadside LiDAR

Compared to the mobile LiDAR models that are developed for an AV to explore and understand its ever-changing surrounding environment, the roadside or stationary LiDAR is mainly used to detect moving objects in a fixed environment, and its point cloud distribution is different from that of mobile LiDAR [15]. In a roadside LiDAR point cloud, vehicles are distributed randomly within the LiDAR scanning range, and their point features (e.g., density and shape) change dynamically in space, while static backgrounds, such as buildings, are invariant [26].

Based on these features, the first step in vehicle detection and tracking using roadside LiDAR is to perform background filtering to remove static objects such as buildings and trees [27]. The next step is to perform object classification to classify the road users into pedestrians, bicycles, and vehicles [8,10]. Finally, the vehicle is tracked to monitor its trajectory and status [9,28]. However, the roadside object detection methods described above are mainly based on point cloud features, and their algorithms run slowly. Some studies do not even report the running time of the proposed algorithms [8,10].

Object tracking algorithms based on roadside sensors include single object tracking and multi-object tracking [29]. Among them, single-object tracking refers to tracking only one object in each frame, which has the advantages of high accuracy and accurate detection. However, in the actual transportation scene, LiDAR often detects multiple vehicles, which greatly reduces the usefulness of single-object tracking. The use of multi-object tracking (MOT) algorithms is necessary [30]. MOT is used to track multiple objects in the same data frame and assign different IDs to all the objects to obtain the trajectory of each ID. The first step in MOT is object matching, i.e., solving for the optimal matching results of all the objects in two consecutive frames before and after. The Hungarian algorithm is a combinatorial optimization algorithm for solving task assignment problems for the matching of bipartite graphs [21]. The traditional Hungarian algorithm considers only the Euclidean distance between object centroids when performing multi-object matching. Although this method is fast in finding solutions, it only selects the Euclidean distance between center points as the efficiency matrix and considers a single factor [31]. When an object is temporarily occluded, it is recognized as two different objects, which leads to tracking errors of the same object in different frames and greatly reduces the tracking performance.

Therefore, this paper proposes to borrow the deep learning-based object detection algorithm used in in-vehicle LiDAR and apply it to the roadside so that it can process the point cloud data at a full frame rate without sacrificing the accuracy. The tail delay is less than 100 ms, and the traditional Hungarian algorithm is improved to solve the short-term occlusion problem and repair the trajectory during the occlusion period while ensuring the running speed of the algorithm.

## 3. Methodology

### 3.1. Vehicle Detection Based on ResNet18

In this paper, we propose a vehicle detection architecture that takes 3D point clouds as the input and predicts the full 3D bounding box. The architecture is an end-to-end ResNet18-based deep learning model. The main process of the model architecture is as follows:

Step 1: The original point cloud is mapped into a BEV and then input into the ResNet18 network. ResNet18 is used as a feature extractor in the architecture to extract the high-level information of the vehicles in the BEV and finally obtain the feature map.

Step 2: The feature maps extracted using ResNet18 are input into 2× deconv branch and 4× deconv branch architecture.

Step 2.1: In the 2× deconv branch, firstly, up-sampling work is done using bilinear interpolation to interpolate each pixel of the feature map to increase its size. This step is performed to obtain a higher resolution feature map to better capture the feature map details and contextual information. This feature map is then input to the objectness classifier and the 3D box regressor. The task of the objectness classifier is to determine whether each region in the feature map contains a vehicle or not, and the output of this branch is typically a binary categorized probability value indicating presence or absence. The task of the 3D box regressor is to predict the position and size information of the 3D bounding box of each detected vehicle. Finally, the outputs of the objectness classifier and 3D box regressor are combined and merged into 3D proposals, and the 3D proposals information is back-projected into the BEV via BEV proposals.

Step 2.2: In the 4× deconv branch, the same up-sampling operation is performed to further increase the spatial resolution of the feature map. Then, the feature map is connected to the region of interest (ROI) pooling operation. ROI pooling is used to extract the features in the ROI region from the feature map based on the proposed bounding box for further processing.

Step 3: After the output of ROI pooling is merged with the BEV proposals, it is connected to a 3D box regressor through a multilayer perceptron (MLP), which is used to further refine and predict the 3D bounding box information of the vehicle.

Step 4: Finally, the model outputs the vehicle detection results, including the 3D bounding box information and the probability of the presence of an object.

The entire architecture is shown in Figure 1.

#### 3.1.1. BEV Transformation

In vehicle detection scenes, most of the vehicles to be detected are on the surface of the road. Compared with the front-view image captured by the camera, objects in the point cloud in the BEV are at various locations in the same plane. They do not overlap with each other; thus, the detection accuracy is higher. The traditional point cloud BEV mapping process directly maps the 3D point clouds to the ground, and the point cloud height information is lost. In order to retain the height information in the BEV mapping process, borrowing an idea from the VoxelNet [18] method, the 3D point cloud is divided into K slices and compressed along the Z-axis direction to map them into a BEV containing height information. The improved BEV mapping is shown in Figure 2, and the specific steps are described as follows:

Step 1: Input the raw point cloud data and discretize them using a 2D grid with a resolution of 0.1 m.

Step 2: To encode more detailed height information, the point cloud is divided into K slices according to height. To prevent the presence of objects beyond the roadway (overpasses, tree canopies, etc.) during BEV transitions, we limit the height to [−h, −h + 3], where h is the installation height of LiDAR and 3m is the height of truck-type vehicle. In the actual application process, readers can change it according to their needs.

Step 3: Determine whether a point cloud exists within each grid cell.

Step 4: For each grid cell in which a point cloud exists, the height $C_{i,j,k}^{H}$ is the maximum height of the points within the cell, the intensity $C_{i,j,k}^{I}$ is the intensity of the point with the maximum height, and the density $C_{i,j,k}^{D}$ is the number of points contained within each cell.

Step 5: Output the mapped BEV point cloud map with the encoding information.

#### 3.1.2. ResNet18-Based Feature Extraction

Deep convolutional neural networks (DCNNs) fuse features at various levels, such as local features and global features, and thus can obtain richer features by increasing the number of layers in the network [32]. However, if the network layers are too deep, the traditional convolutional neural network (CNN) suffers from the degradation problem, i.e., when the number of network layers reaches a certain level and is too complex, the accuracy of the algorithm will saturate and stabilize for a period of time, followed by a rapid decline. Therefore, He et al. [33] proposed the residual network (ResNet) in order to optimize the number of network layers. ResNet turns the original network of several layers of the original DCNN into a residual recognition block, which can ensure the same input and output in the feature extraction process. In order to speed up the operation and improve the object detection accuracy, this paper uses ResNet18 as the network for feature extraction. The network has a total of 18 hidden layers, including 17 convolutional layers and 1 fully connected layer, and the network structure is shown in Figure 3.

#### 3.1.3. 3D Proposals Network

The 3D proposals network (RPN) has become the key component of 3D object detectors [34]. Given a feature map, the network generates 3D bounding box proposals from a set of 3D prior boxes. The parameters of each 3D bounding box include (x, y, z, l, w, h, θ), which are the center, size, and navigation angle of the 3D bounding box in the LiDAR coordinate system. In this paper, we use the following three strategies in the 3D proposals network:

(1) In order to make the training of proposal regressions easier, the navigation angle of the 3D bounding box is constrained to [0-degree, 90-degree], which is close to the actual navigation angle of most road scene objects.

(2) For ultra-small objects at a 0.1 m resolution, 2× up-sampling is performed on the feature maps extracted using ResNet18.

(3) The sparsity of the point cloud leads to many empty anchors. In order to reduce the computational effort, we remove all the empty anchors during the training and testing process.

#### 3.1.4. Objective Optimization and Network Training

For the constructed vehicle detection model, back propagation is performed using stochastic gradient descent to update the weights and biases of the network, as shown in Equation (1).
(1)v=βv−α∇ωx←x+v
where α denotes the learning rate and v denotes the momentum factor. When the direction of the negative gradient is the same as the direction of v, it indicates that the update is in the right direction.

The loss function of the model mainly includes a loss of classification and a loss of the 3D bounding box estimation, and the loss function is weighted to obtain the total loss function as follows:(2)$$L_{total}=\mu_{1}L_{object}+\mu_{2}L_{box}$$
where $\mu_{1}$ and $\mu_{2}$ are the weighting coefficients.

For the object classification loss, the calculation is performed using binary cross-entropy, as shown in Equation (3):(3)$$L_{object}=N^{p}\times logN^{t}-(1-N^{p})log(1-N^{t})$$
where $N^{p}$ and $N^{t}$ are the predicted and true values of the object category, respectively.

For the location regression loss, the commonly used loss function is the intersection over union (IoU) loss function, which reflects the degree of overlap between the 3D bounding box predicted by the model and the real 3D bounding box, as shown in Figure 4. The yellow 3D bounding box is the real bounding box, and the green 3D bounding box is the model-predicted bounding box. Then, the IoU of the predicted bounding box and the real bounding box can be calculated using Equation (4).
(4)$$IoU=\frac{\left|True3Dbox\cap Pre3Dbox\right|}{\left|True3Dbox\cup Pre3Dbox\right|}$$

However, there are some limitations in Equation (4). If the prediction bounding box and the real bounding box do not intersect, then IoU = 0 and the gradient function is 0 at this point; thus, the network cannot learn and update the parameters. There may also be a problem when IoU is equal but does not reflect the prediction bounding box direction, shape, and size, as shown in Figure 4a,b. Therefore, in this paper, the IoU loss function is improved by considering not only the overlapping volume between the prediction bounding box and the real bounding box, but also the degree of similarity between the two boxes. The improved IoU loss function is shown in Equation (5):(5)$$L_{box}=1+\lambda \Theta +\frac{\vartheta (C^{p},C^{t})}{l}-IoU$$
where λ is the weight coefficient, calculated as shown in Equation (6); Θ is the similarity index of the prediction bounding box and the real bounding box, calculated as shown in Equation (7); $C^{p}$ and $C^{t}$ are the center coordinates of the prediction bounding box and the real bounding box, respectively; ϑ is the Euclidean distance between the center coordinate points of the prediction bounding box and the real bounding box; and l denotes the longest distance between the prediction bounding box and the real bounding box enclosing the minimum convex envelope domain.
(6)$$\lambda =\frac{\Theta }{\Theta +1-IoU}$$
(7)$$\Theta =\frac{4}{\pi^{2}}({\left(arctan\frac{w^{p}}{l^{p}}-arctan\frac{w^{t}}{l^{t}}\right)}^{2}+{\left(arctan\frac{l^{p}}{h^{p}}-arctan\frac{l^{t}}{h^{t}}\right)}^{2}\;+{\left(arctan\frac{h^{p}}{w^{p}}-arctan\frac{h^{t}}{w^{t}}\right)}^{2})$$
where $l^{p}$, $w^{p}$, and $h^{p}$ are the length, width, and height of the predicted frame, respectively, and $l^{t}$, $w^{t}$, and $h^{t}$ are the length, width, and height of the real frame, respectively.

### 3.2. Vehicle Tracking at Three Stages

After the completion of the vehicle detection phase, we will realize the vehicle tracking in three steps to recover the complete trajectory of the vehicle in the LiDAR detection area and provide data support for the subsequent traffic flow state sensing and connected-AV applications. The framework of the vehicle tracking algorithm in this paper is shown in Figure 5, and the specific steps are described as follows:

Step 1: The vehicle detection results of two consecutive frames are classified into three cases, and global object matching is performed using the improved Hungarian algorithm.

Step 2: Due to the existence of the occlusion problem, a single vehicle may be identified as different objects before and after the occlusion in the LiDAR detection area. We propose time–space logicality and trajectory similarity to determine whether the objects before and after the occlusion are the same object and to merge the trajectories of the same object.

Step 3: After the trajectories of the same object are merged, the trajectories of the object during the occlusion period are still missing; therefore, a linear interpolation algorithm is used in this paper to recover these missing trajectories.

#### 3.2.1. Object Matching for Consecutive Frames Based on the Improved Hungarian Algorithm

In order to solve this problem, this paper adds a new object IOU feature by considering the Euclidean distance between object centroids. Since urban traffic is often limited to 50 km/h and LiDAR’s single-frame scanning time is 0.1 s, the distance traveled by a vehicle in the one-frame time interval is approximately equal to 1.3 m. The length of a small vehicle is certainly greater than 1.3 m, so the same object is certain to intersect in two consecutive frames of the point cloud.

In this way, the spatial position and spatial volume of the objects in the front and back frames can be incorporated into the efficiency matrix at the same time when judging the similarity between the front and back frames. Assuming that m objects are detected in the previous frame and n objects are detected in the current frame, the efficiency matrix is as follows:(8)$${\left(C_{ij}\right)}_{m\times n}=\left\{\begin{array}{cc} \begin{array}{cc} C_{11} & C_{12}\\ C_{21} & C_{22} \end{array} & \begin{array}{cc} \cdots & C_{1n}\\ \cdots & C_{2n} \end{array}\\ \begin{array}{cc} \vdots & \vdots \\ C_{m1} & C_{m2} \end{array} & \begin{array}{cc} \ddots & \vdots \\ \cdots & C_{mn} \end{array} \end{array}\right\}$$
where $C_{ij}$ is the correlation between the *i*th object in the previous frame and the *j*th object in the current frame, and a larger value indicates that it may be the same object, which is calculated as:(9)$$C_{ij}=\alpha C_{IoU}\left(i,j\right)+\frac{\beta }{C_{d}(i,j)}$$
where α and β are the normalized weight coefficients and $C_{IoU}\left(i,j\right)$ is the IoU of the *i*th object in the previous frame to the *j*th object in the current frame. $C_{d}(i,j)$ is the Euclidean distance between the *i*th and *j*th objects. In this paper, we treat the two metrics, $C_{IoU}\left(i,j\right)$ and $C_{d}(i,j)$, as are equally important; therefore, we set α=0.5 and β=0.5. In real-scenario applications, readers can re-assign the values according to the importance of the two metrics.

By using the improved Hungarian matching method, the objects of two consecutive front and back frames are linked together. In order to form a vehicle trajectory with multiple object vehicle positions, this paper proposes a multi-object matching framework based on the improved Hungarian algorithm, as shown in Algorithm 1. It only needs to separately match multiple sets of before and after frames to obtain the vehicle trajectories of multiple frames.
**Algorithm** **1**: Object matching pseudo-codeInput: All_object, current_frame_object, previous_frame_objectOutput: All_object
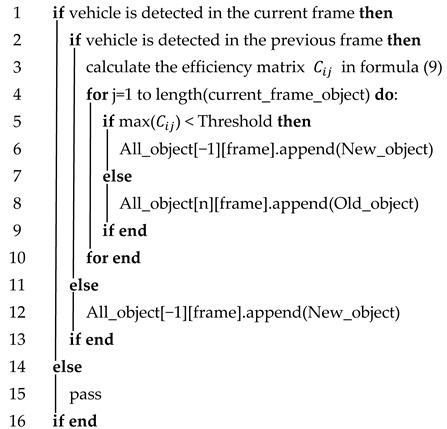


In Algorithm 1, the input is the All_Object matrix (the row is the number of currently matched objects, and each row represents one object. The column is the number of currently matched frames, and each column represents the coordinate points of the object’s attributes in that frame. The matrix of objective attributes detected in the previous and current frames are previous_frame_object and current_frame_object, respectively, and the rows of the matrix are the number of objects detected in that frame. The columns are the attributes [Type,x,y,l,w,θ] of each object. New_object indicates that the vehicle detected in the current frame has not appeared in the previous frame. Conversely, Old_object indicates that the vehicle detected in the current frame has appeared in the previous frame. The framework divides the objects detected in the before and after frames into three cases:

Case 1: If no object is detected in the current frame, no operation is performed, as shown in Figure 6a.

Case 2: If one object is detected in the current frame and no object is detected in the previous frame, all the objects detected in the current frame are New_objects, and the attributes of the New_objects are appended to All_Object, as shown in Figure 6b.

Case 3: If objects are detected in both the front and back frames, the efficiency matrix $C_{ij}$ is calculated. Each object *j* is looped in the current frame, and their correlations are judged using all objects *i* in the previous frame. If the maximum correlation is still less than the set threshold, the *j*th object is considered as a New_object; otherwise, the *j*th object is the Old_object, as shown in Figure 6c.

#### 3.2.2. Object Merger in Fixed Time Intervals

Occlusion problems may occur during vehicle detection, which causes interruptions in object matching. In Figure 7, since object ID4 in frame T obscures object ID3, object ID3 in frame T − 1 cannot be matched in frame T, resulting in a missing object. In frame T + 1, object ID3, which is blocked by object ID4, will be identified as a new object, i.e., object ID5, by the matching algorithm. As a result, Object ID5 in frame T + 1 cannot find the corresponding matching object in frame T. Assuming that M objects are detected in time interval N, the spatial–temporal diagram they form is shown in Figure 8. The function of object merging in this section is to identify objects (e.g., ID3 and ID5) that may have trajectory interruption phenomena and merge them into one object.

To achieve object trajectory merging, this paper proposes a merging algorithm based on time–space logicality and trajectory similarity; see Algorithm 2. In Algorithm 2, if the number of frames detected in the detection interval for $n_{1}$ is less than a certain threshold, the object is considered to have possible trajectory interruptions. If two (or more) objects satisfy both time–space logicality and trajectory similarity, they are considered to be the same object and need to be merged.
**Algorithm** **2**: Merging of objects with interrupted trajectories.**Input**: All_Object**Output**: Merger_ Object (Object matrix that can be merged)
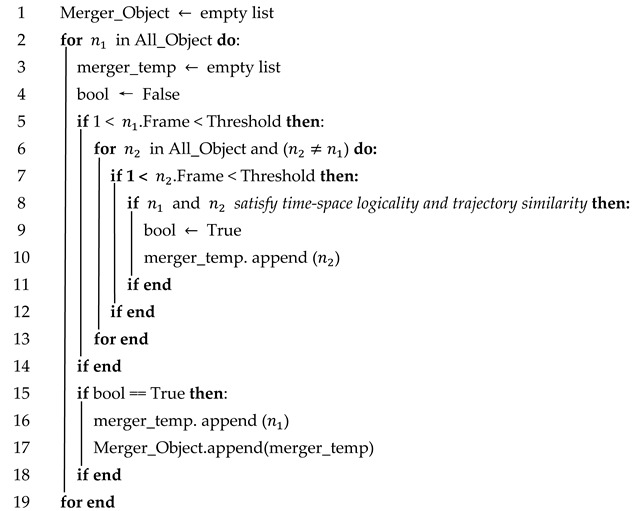


In Algorithm 2, the threshold is calculated as follows:(10)$$Threshold=int\left(\frac{\max\left(Y_{BEV}\right)-\min\left(Y_{BEV}\right)}{19.44}\times 10\right)$$
where max($Y_{BEV}$) and min($Y_{BEV}$) denote the maximum and minimum values of the Y-axis in the BEV mapping, respectively, and int() is a reserved integer operator.

In this paper, we crop the BEV dimension of the y-axis to [−100 m, 100 m] so that the traveling distance of the vehicle in the detection area is 200 m. In urban roads, the maximum speed of the vehicle is 50 km/s ≈ 13.89 m/s; therefore, the traveling time of the vehicle in the detection area is 200/13.89 ≈ 14 s. The interval between each frame of the point cloud data is 0.1s. In the case of no occlusion, if the vehicle’s consecutive trajectory is less than 140 frames, the vehicle is considered to be occluded in the detection area, resulting in an interrupted trajectory. Therefore, in this paper, the threshold in Algorithm 2 is set to 140.

$n_{1}$ and $n_{2}$ are considered to satisfy time–space logicality and trajectory similarity if any of the following four cases are satisfied, as shown in Figure 9 and Algorithm 3.

Case 1: If both $n_{1}$ and $n_{2}$ are traveling in the positive direction of the x-axis, the x-coordinate where the first frame of $n_{1}$ is detected is greater than the x-coordinate where the last frame of $n_{2}$ is detected, and the time when $n_{1}$ is detected later than the time when $n_{2}$ disappears.

Case 2: If both $n_{1}$ and $n_{2}$ are traveling in the positive direction of the x-axis, the x-coordinate of the last detected frame of $n_{1}$ is smaller than the x-coordinate of the first detected frame of $n_{2}$, and the time when $n_{1}$ disappears is earlier than the time when $n_{2}$ is detected.

Case 3: If both $n_{1}$ and $n_{2}$ are traveling in the negative direction of the x-axis, the x-coordinate of the first detected frame of $n_{1}$ is smaller than the x-coordinate of the last detected frame of $n_{2}$, and the time when $n_{1}$ is detected is later than the time when $n_{2}$ disappears.

Case 4: If both $n_{1}$ and $n_{2}$ are traveling in the negative direction of the x-axis, the x-coordinate of the last detected frame of $n_{1}$ is larger than the x-coordinate of the first detected frame of $n_{2}$, and the time when $n_{1}$ disappears is earlier than the time when $n_{2}$ is detected.

Each time–space logicality case corresponds to the trajectory similarity. Take case 1 as an example: n2′ is the predicted trajectory of n2, and the predicted time is [n2.end_time, n1.start_time]. The Euclidean distance between the xy coordinate of the last frame of n2′ and the xy coordinate of the first frame of n1 is judged, and if it is less than the distance threshold, then n1 and n2 are considered to satisfy the trajectory similarity, as shown in Figure 10.
**Algorithm** **3**: Whether n1 and n2 satisfy time-space logicality and trajectory similarity**Input**: Attributes of vehicle n1. (n1.start_x, n1.start_time, n1.end_x, n1.end_time);Attributes of vehicle n2. (n2.start_x, n2.start_time, n2.end_x, n2.end_time)**Output**: bool (Whether n1 and n2 satisfy space-time logicality and trajectory similarity)
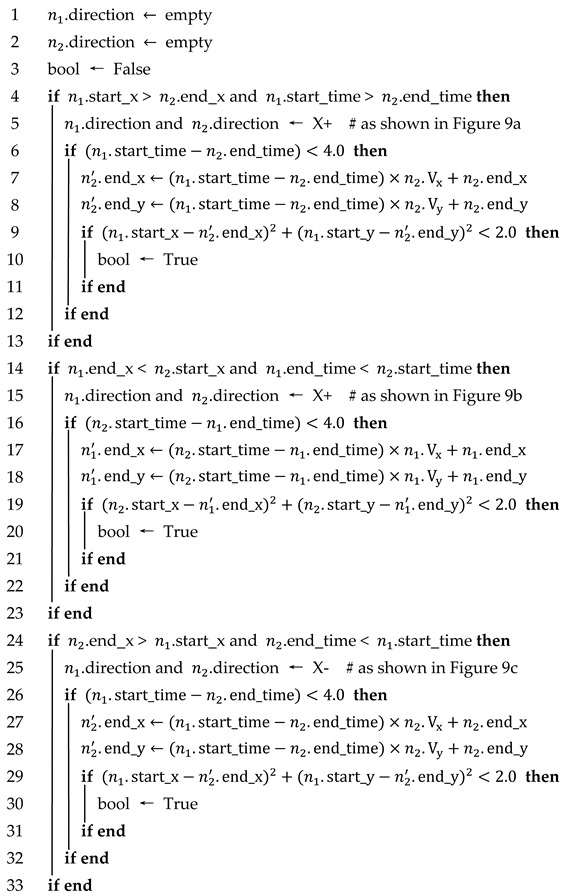

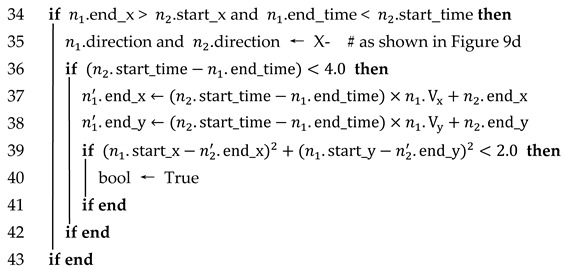


In Algorithm 3, $V_{x}$ and $V_{y}$ are the average velocity components of $n_{1}/n_{2}$ in the x and y directions, respectively. In order to reduce the computational pressure, we set the maximum prediction frame to 40 frames; that is, when the trajectory of a vehicle (ID = i) is interrupted, its trajectory is predicted, and if no similar vehicle is matched in 40 consecutive frames of prediction, it is considered that the vehicle has already moved out of the detection area. In this case, the vehicle’s trajectory in the detection area is outputted as All_Object[i][start.frame: end. frame]. Otherwise, if another object or objects are matched within 40 consecutive frames, they are merged into one object and the missing trajectories are completed.

Additionally, we set the Euclidean distance threshold to 2.0 m, which is an empirical value and does not have a formula. Although some scholars use 1.5 m as the Euclidean distance threshold between predicted and real coordinates, their maximum prediction interval is 30 frames because the sensor they used was the VLP-16, which has a shorter detection distance [10].

#### 3.2.3. Completion of Missing Trajectories

When we finish the object merging, there are still missing trajectories for the merged objects, as shown in Figure 11. In Figure 11, $n_{1}$ and $n_{2}$ were previously identified as two different objects, and after using the object merging algorithm, both were merged into the same object ID3. In order to complete the missing trajectories in the merged object, a linear interpolation-based trajectory completion algorithm is proposed in this paper; see Algorithm 4. The algorithm can automatically find the coordinates and frame number of each missing trajectory and use the linear interpolation method to complete the missing trajectories, finally returning the complete trajectory of the vehicle in the detection area.
**Algorithm** **4**: Complementary algorithm for missing trajectoriesInput: The object containing the missing trajectory, in the form of a list, with the value of None for the missing trajectory (T_miss)Output: Object after trajectory completion (T)
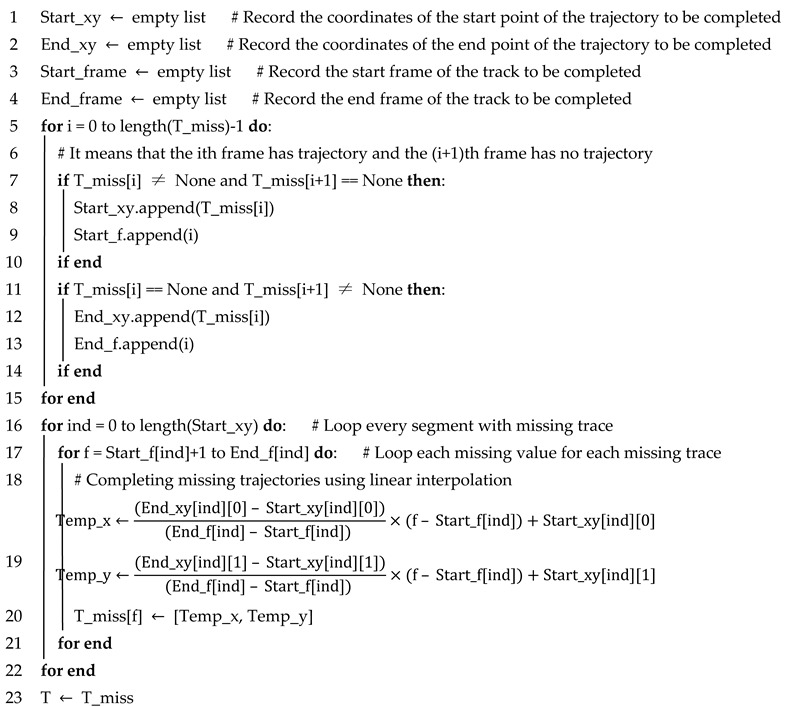


## 4. Case Study

### 4.1. Experimental Preparation

#### 4.1.1. Experimental Dataset Collection

Conducting roadside LiDAR experiments in real road scenarios may have an impact on the personal safety of the experimenters, especially during the algorithm testing phase. Therefore, the experimental data in this paper are derived from the mature project KITTI dataset and MATLAB/Simulink simulation data.

(1)KITTI dataset

The point cloud dataset_1 used herein is derived from the KITTI benchmark dataset [11], which was acquired using a Velodyne HDL-64E model LiDAR in a real environment. This LiDAR was installed at a height of 1.73 m above ground level, and the data format of each frame of the collected point cloud has a matrix of n rows (3D spatial points) and four columns (x, y, z, and intensity). The corresponding labels of this matrix are m rows (m objects) and eight columns (object class, xyz coordinate of object center point, length, width, and height of the object, and object navigation angle). The dataset has a total of 7481 frames of training samples and 7518 frames of test samples. The training samples are split into the training and validation sets in a ratio of 4 to 1.

The point cloud data used for object tracking is from a static scene of an intersection in the KITTI dataset. We used the data collected in this scene to evaluate the roadside LiDAR object tracking algorithm.

(2)MATLAB/Simulink dataset

The point cloud dataset_2 used herein is derived from the MATLAB/Simulink simulation data. Conducting real road experiments may require expensive equipment, labor, and time costs, compared to simulation experiments, which are more cost-effective and can save significant resources [35]. The simulation steps based on MATLAB/Simulink are as follows:

Step 1: Use UE (unreal engine) to build the simulation environment and generate the AutoVrtlEnv.uproject project file.

Step 2: Create a vehicle trajectory in the simulation environment and save it as a trajectory class containing xy coordinates and the yaw angle.

Step 3: Build the MATLAB/Simulink simulation model as shown in Figure 12. The model contains the following modules:

Modul 1: Simulation 3D scene configuration. This module reads the AutoVrtlEnv.uproject project file as a simulation scenario.

Modul 2: LiDAR sensor configuration. This module is used to configure the LiDAR’s parameters, position, and pose. In this simulation experiment, the LiDAR parameters are set to model HDL-64E and the installation height is 1.73 m above the ground, which is consistent with the LiDAR Sensor in the KITTI dataset.

Modul 3: Vehicle trajectory configuration. This module reads the trajectory of the vehicles and is used to control the time, position, and yaw angle at which each vehicle appears.

Modul 4: Point cloud data collection and visualization. This module is used to collect and visualize the point cloud data detected by the LiDAR sensors. The point cloud data format is aligned with KITTI, and the point cloud is stored as a .csv file for each frame.

Finally, we used the MATLAB/Lidar Labeler application to interactively label real data in the point clouds and compare the results with the Python environment for object detection and tracking.

#### 4.1.2. Experimental Environment Configuration

The PC configuration was an AMD Ryzen 7 3700X eight-core Processor CPU@3.6GHz, DDR4 32G memory, GeForce RTX 3060 GPU (12G memory). The operating system was Win10, the code running environment was Python 3.6. The deep learning framework was Torch 1.5, and CUDA10.2 was used for GPU acceleration. The parameters used in the deep learning training process for object detection are shown in Table 1.

### 4.2. Experimental Results and Evaluation

The experimental results of each step are shown in Figure 13. The original point cloud is input, then BEV mapping is performed. Object detection is carried out according to the framework of the vehicle detection algorithm proposed in this paper, and the vehicle detection results of each frame are derived. Finally, the detected vehicles are tracked to obtain their running trajectories. As an example, a demo video of the tracking results of the KITTI dataset can be viewed on YouTube at the following link: https://youtu.be/GfqKXwJeDBU, accessed on 26 September 2023.

#### 4.2.1. Vehicle Detection

In order to evaluate the performance of the proposed vehicle detection algorithm, classical object detection algorithms (PointNet++ [17] and VoxelNet [18]) are used for comparison. The detection results using the KITTI dataset and MATLAB/Simulink simulation dataset were counted separately, as shown in Table 2, where TP (true positive) indicates that the algorithm correctly detected the vehicle and the vehicle also exists in the actual scenario. FP (false positive) indicates that the algorithm incorrectly predicts a sample that does not exist in the scenario to be a vehicle; that is, misdetection. FN (false negative) indicates that the algorithm does not detect the vehicle that exists in the scenario; that is, a missed detection.

Based on the vehicle detection statistics shown in Table 2, the metrics commonly used in object detection algorithms such as precision, recall, F1-score, and FPS (frames per second) are used to evaluate the vehicle detection algorithm proposed in this paper. The evaluation results are shown in Table 3.

(1)Precision:

Precision, also known as the positive predictive value (PPV), measures the accuracy of positive predictions made by the detection model. It is calculated as the ratio of true positives (TP) to the sum of true positives and false positives (FP). It is calculated as shown in Formula (11).
(11)$$Pr=\frac{TP}{TP+FP}$$

(2)Recall:

Recall, also known as sensitivity or the true positive rate (TPR), measures the ability of the model to correctly detect positive samples (objects) from all the actual positive samples in the dataset. It is calculated as the ratio of true positives (TP) to the sum of true positives and false negatives (FN). It is calculated as shown in Formula (12).
(12)$$Re=\frac{TP}{TP+FN}$$

(3)F1-score:

The F1-score is the harmonic mean of precision and recall and provides a single value that balances both metrics. It is useful when there is an uneven class distribution between positive and negative samples. The F1-score has a value interval of [0–1], and the closer to it is to 1, the better the performance of the vehicle detection model. It is calculated as shown in Formula (13).
(13)$$F1-score=\frac{2*Pr*Re}{Pr+Re}$$

(4)FPS:

FPS (frames per second) indicates the number of frames per second of the point cloud that the model can process in vehicle detection. It is a measure of the real-time performance of the model. A high FPS means that the model is able to process the point cloud quickly and performs well in real-time applications. A lower FPS value may result in insufficient processing speeds for real-time applications. FPS is affected by a number of factors, including model complexity, hardware capabilities, and the number of point clouds.

As can be seen from Table 3, when used on both the KITTI dataset and the MATLAB/Simulink simulation dataset, our algorithm achieves the optimum in all four metrics, with F1-scores of 96.97% and 98.58%, respectively, and the FPS is able to achieve real-time vehicle detection. In addition, no matter which algorithm is used, the detection effect using the MATLAB/Simulink simulation dataset is better than that of the KITTI dataset. The reason for this may be that the simulation environment is more ideal, and the LiDAR point cloud distribution is not affected by the external environment (e.g., LiDAR does not shake, and the launched laser line returns 100% when it encounters an obstacle).

#### 4.2.2. Vehicle Tracking

In order to verify the effectiveness of the three-stage vehicle tracking algorithm proposed in this paper, we compared it with the traditional Hungarian object tracking algorithm using the two datasets, as shown in Figure 14 and Figure 15. In Figure 14 and Figure 15, different colors indicate the trajectories of vehicles with different IDs tracked by the algorithm.

By comparing Figure 14 and Figure 15, it can be found that the three-stage vehicle tracking algorithm proposed in this paper is more effective and can solve the short-time occlusion problem during the tracking process, preventing the interruption of the tracked vehicles. In addition to comparing the visualization results, the CLEAR MOT [36], which is the common performance evaluation metric for MOT, was used as the benchmark, including multiple object tracking accuracy (MOTA), multiple object tracking precision (MOTP), ID switch (IDSW), ID-F1, and the average time of a single frame (ATSF). An explanation of each metric and its calculation formula are given below.

(1)MOTA:

MOTA gives an intuitive measure of the tracker’s performance in detecting objects (FN, FP) and maintaining trajectories (IDSW). The value of MOTA may be negative, but the closer it is to 1, the better the performance of the tracker. MOTA is averaged by summing the metrics across all the frames rather than counting one MOTA per frame and then averaging them. The formula is as follows:(14)$$MOTA=1-\frac{\sum_{1}^{length(Frames)}\left({FN}_{f}+{FP}_{f}+{IDSW}_{f}\right)}{\sum_{1}^{length(Frames)}{GT}_{f}}$$
where ${GT}_{f}$ denotes the ground truth labeling of the *f*-th frame.

(2)MOTP:

MOTP primarily measures the accuracy of the tracking and localization distance. The distance metric used in this paper is the Euclidean distance. The closer it is to 0, the better the accuracy.
(15)$$MOTA=\frac{\sum_{1}^{length(Frames)}\sum_{1}^{length(objects)}d_{f,i}}{\sum_{1}^{length(Frames)}c_{f}}$$
where $d_{f,i}$ is the Euclidean distance between the i-th vehicle of the f-th frame and the corresponding ground truth, and $c_{f}$ is the number of successful matches in the f-th frame.

(3)IDSW:

IDSW is the total number of times the ground truth for the same object within the dataset was matched to an ID that was switched. The smaller the IDSW, the better the matching.

(4)ID-F1:

Similar to the F1-Score metric for vehicle detection, this metric represents the harmonic mean of the accuracy and recall of vehicle tracking. This metric is more responsive to tracking capabilities than MOTA, which only considers IDSW, because it takes ID into account. ID-F1 takes on a value within the range of [0, 1], and the closer it is to 1, the better the accuracy and recall. It is calculated as follows:(16)$$ID-F1=\frac{2{ID}_{TP}}{2{ID}_{TP}+{ID}_{FP}+{ID}_{FN}}$$

(5)ATSF:

ATSF is the average single-frame runtime of the tracking algorithm, and the unit is seconds.

A comparison of these metrics is presented in Table 4.

Table 4 shows that the algorithm proposed in this paper has a significant improvement in the MOTA and ID-F1, which is due to the work shown in Section 3.2.2 and Section 3.2.3 that reduces the IDSW and increases the trajectory integrity. Since the method proposed in this paper has three phases of operation, the detection time is increased; however, its ATSF is still sufficient for real-time tracking of vehicle trajectories. In addition, since the method proposed in this paper predicts the missing trajectory and uses it as the detection trajectory, it results in a slightly higher MOTP than the conventional method due to the prediction error.

## 5. Conclusions and Discussion

### 5.1. Conclusions

In this paper, we have developed and optimized algorithms that enable roadside infrastructure equipped with LiDAR to detect and track vehicles effectively in real-time. Our primary contributions include a ResNet18-based vehicle detection framework and a novel three-stage vehicle tracking algorithm. These methods significantly enhance real-time vehicle detection and tracking, outperforming other methods in computational speed and making them suitable for real-time applications.

Our vehicle detection architecture, which improves BEV mapping and the optimizer’s loss function, has demonstrated remarkable results. When tested on the KITTI dataset and the MATLAB/Simulink simulation dataset, our algorithm achieved optimal performance across all metrics. The F1-Score reached 96.97% and 98.58% for the KITTI dataset and the MATLAB/Simulink simulation dataset, respectively, and the frames per second (FPS) were sufficient to achieve real-time vehicle detection.

Furthermore, our proposed three-stage vehicle tracking algorithm shows significant improvement in MOTA, ID-F1. The MOTA of the proposed method in this paper was 88.12% and 90.56% for the different datasets, which was improved by 24.97% and 21.18%, respectively, while the ID-F1 was 95.16% and 96.43%, which was improved by 26.86% and 30.62%, respectively. Although the three-stage operation increases the detection time, the ATSF is still sufficient for real-time vehicle trajectory tracking.

### 5.2. Discussion

(1) The MATLAB/Simulink simulation scenario used in this study was optimal, making our algorithm perform more effectively. In future work, we plan to further optimize the simulation scenario by introducing external disturbances (such as heavy rain, snowstorms, sandstorms, etc.,) to test the stability of our proposed algorithm. In the case of heavy rain, for example, one can add particles that move from the sky to the ground with reflective properties in the UE. The amount of rainfall can be simulated by setting the density of the particles.

(2) The experimental dataset used for this study uses a 64-channel high-density LiDAR from velodyne, which costs more than a 16-channel or 32-channel LiDAR. However, with the development of this technology, the selling price of LiDARs from several LiDAR vendors (e.g., Ouster, Roboscene) continues to decrease. As an example, the single price of the 125-channel LiDAR released in 2020 by Roboscene has been reduced to $ 1898 [37]. The average unit price will also be less expensive if purchased in bulk. While the initial cost of using a 64-channel LiDAR may be higher, its ability to provide higher quality point cloud data and detection performance compared to using a 16-channel or 32-channel LiDAR can result in a larger payback in the long term.

(3) The vehicle detection and tracking algorithms presented in this paper are tested using velodyne’s HDL-64E model LiDAR, but 64-channel LiDARs (e.g., OS1-64, Pandar-64) made by other vendors are also applicable. The dimensionality and quality of the data captured using the different models of 64-channel LiDAR sensors are not significantly different, and the tensor size of the data input to the ResNet18 neural network can be kept consistent when mapped in a BEV.

## Figures and Tables

**Figure 1 sensors-23-08143-f001:**
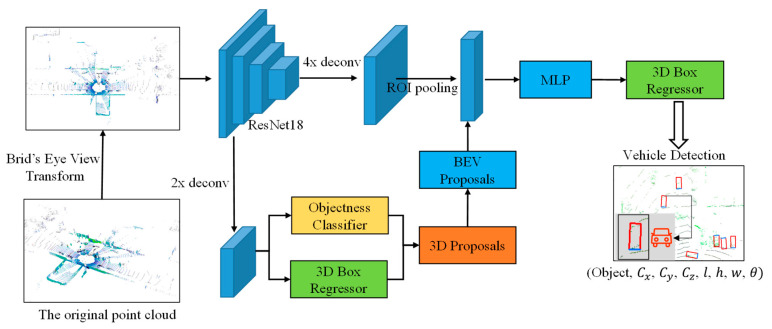
Vehicle detection architecture based on ResNet18.

**Figure 2 sensors-23-08143-f002:**
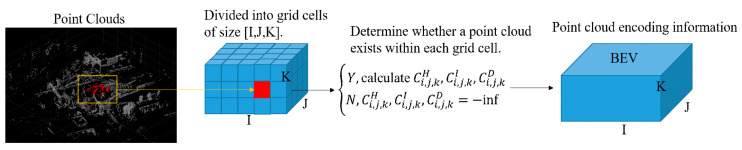
Improved BEV mapping method.

**Figure 3 sensors-23-08143-f003:**
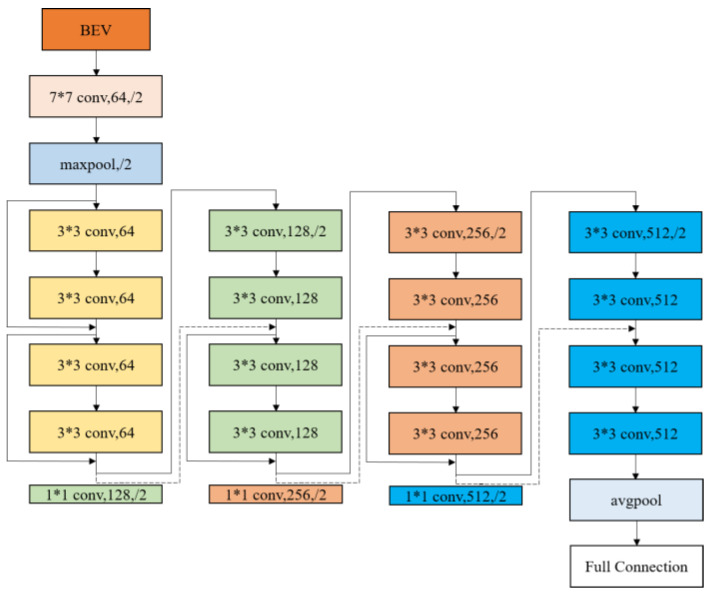
ResNet18 network structure.

**Figure 4 sensors-23-08143-f004:**
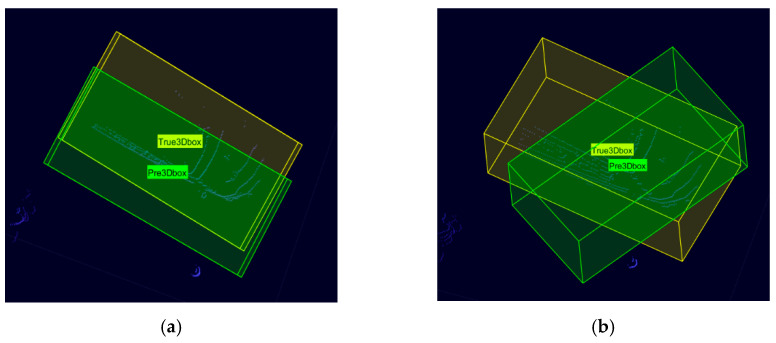
Schematic diagram of the 3D bounding box IoU. (**a**) Scene 1: partial overlap (IoU = 0.56); (**b**) Scene 2: partial overlap (IoU = 0.56).

**Figure 5 sensors-23-08143-f005:**
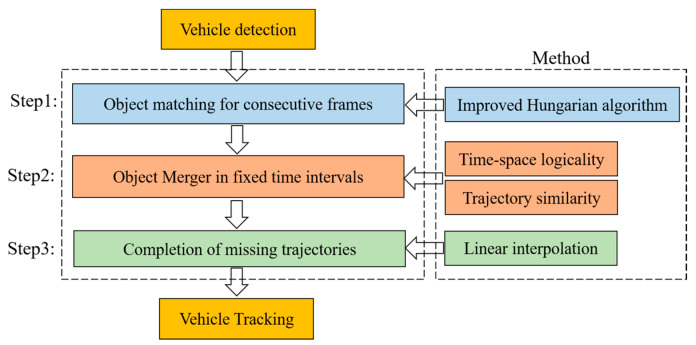
The framework of the vehicle tracking algorithm proposed in this paper.

**Figure 6 sensors-23-08143-f006:**
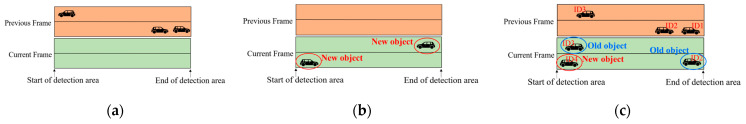
Schematic diagram of the three matching cases. (**a**) Case 1: object not detected in the current frame. (**b**) Case 2: object detected in the current frame, but not in the previous frame. (**c**) Case 3: object is detected in both the previous frame and the current frame.

**Figure 7 sensors-23-08143-f007:**
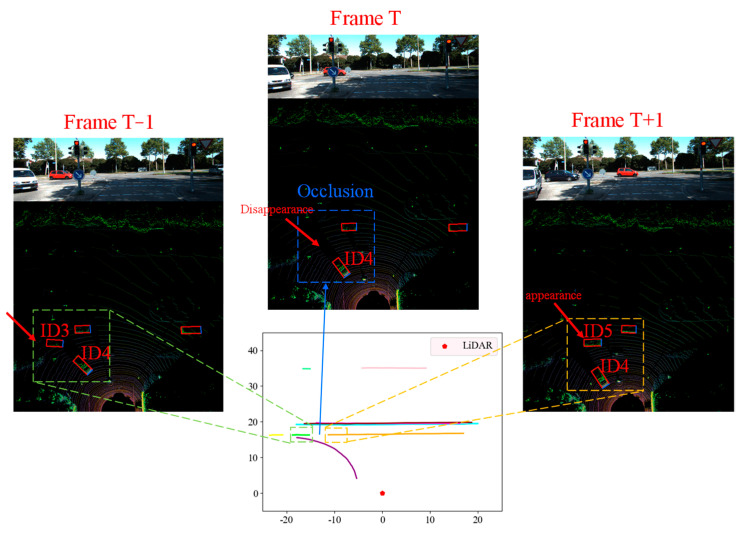
Track interruptions due to occlusion.

**Figure 8 sensors-23-08143-f008:**
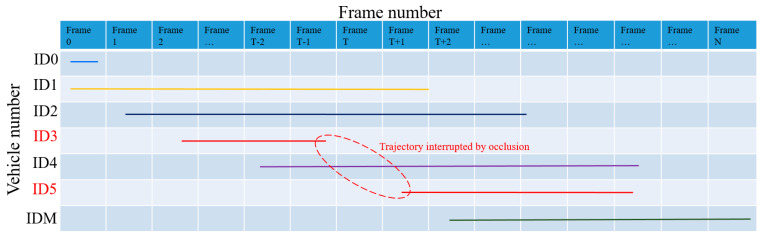
Time–trajectory diagram for vehicle detection.

**Figure 9 sensors-23-08143-f009:**
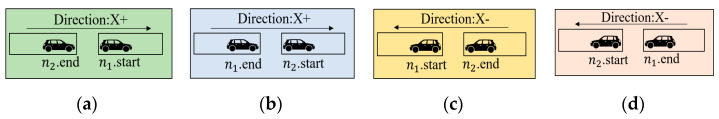
Schematic of space–time logicality. (**a**) Case 1. (**b**) Case 2. (**c**) Case 3. (**d**) Case 4.

**Figure 10 sensors-23-08143-f010:**
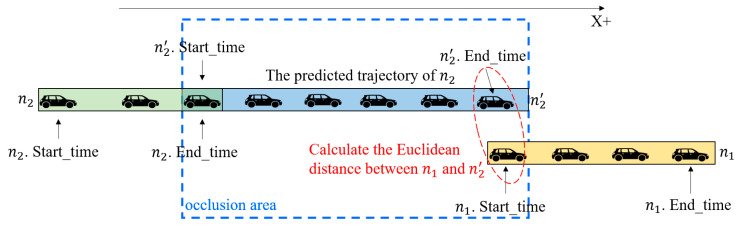
Schematic diagram of the trajectory similarity.

**Figure 11 sensors-23-08143-f011:**
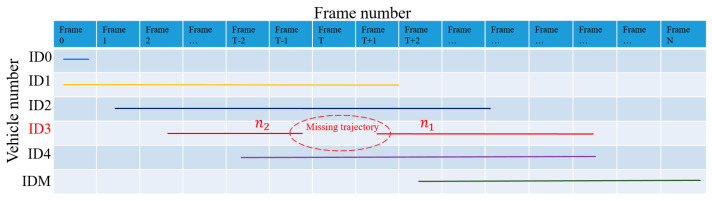
Time–trajectory diagram after object merging.

**Figure 12 sensors-23-08143-f012:**
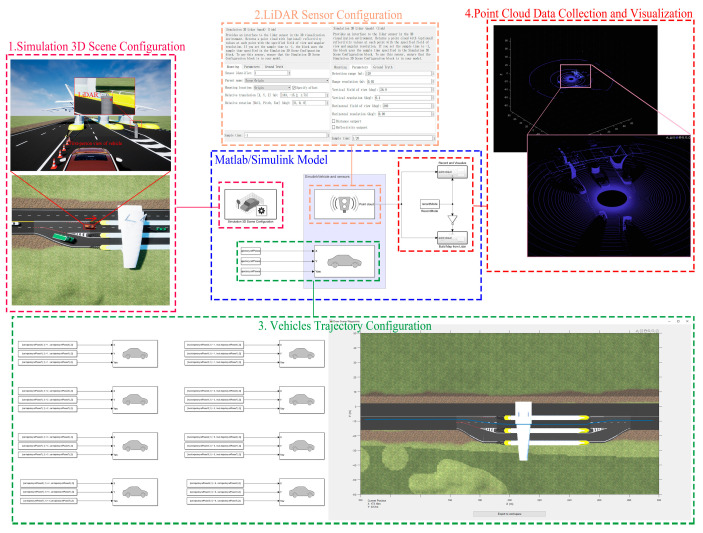
Matlab/Simulink model.

**Figure 13 sensors-23-08143-f013:**
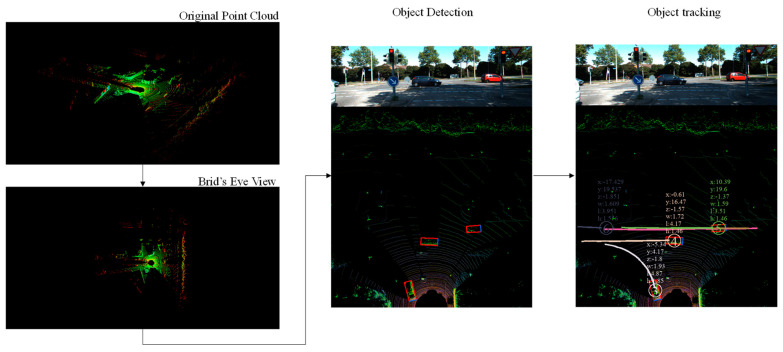
Experimental results for each step.

**Figure 14 sensors-23-08143-f014:**
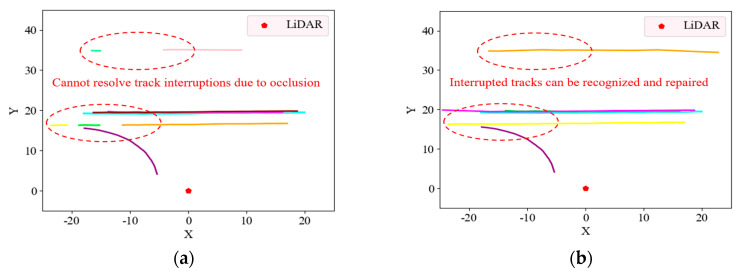
Vehicle tracking trajectories for a given time period in the KITTI dataset. (**a**) Tracking trajectories of the traditional Hungarian algorithm. (**b**) Tracking trajectories of the method proposed in this paper.

**Figure 15 sensors-23-08143-f015:**
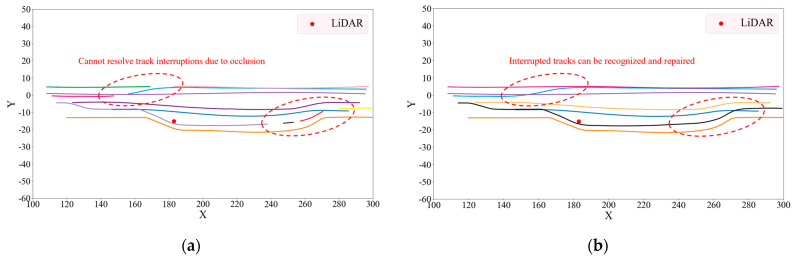
Vehicle tracking trajectories for a given time period in the MATLAB/Simulink simulation dataset. (**a**) Tracking trajectories of the traditional Hungarian algorithm. (**b**) Tracking trajectories of the method proposed in this paper.

**Table 1 sensors-23-08143-t001:** Deep learning hyperparameter configuration.

Parameter	Parameter Value	Parameter Description
Seed	2023	Random seeds used to disrupt the data set and validation set
Epochs	300	Number of times needed to train all samples
Batch_size	16	Batch size of the current node GPU
Lr	$1\times 10^{-3}$	Initial learning rate
Lr_type	cosin	Types of learning rate schedulers
Lr_min	$1\times 10^{-7}$	Minimum learning rate
Optimizer	SGD	Optimizer Type
Checkpoint_freq	5	Breakpoint saving frequency
max_objects	50	Maximum number of detected objects per frame
down_ratio	4	Down-sampling rate to reduce computation while maintaining point cloud features

**Table 2 sensors-23-08143-t002:** Vehicle detection results of different algorithms using different datasets.

Algorithm	KITTI Dataset			MATLAB/Simulink Simulation Dataset		
	TP	FP	FN	TP	FP	FN
PointNet++	536	78	83	483	37	42
VoxelNet	557	49	62	487	33	38
Our method	608	27	11	520	10	5

**Table 3 sensors-23-08143-t003:** Vehicle detection evaluation metrics of different algorithms using different datasets.

Algorithm	KITTI Dataset				MATLAB/Simulink Simulation Dataset			
	Precision	Recall	F1-Score	FPS	Precision	Recall	F1-Score	FPS
PointNet++	87.29%	86.59%	86.94%	2.1	92.88%	92.00%	92.44%	2.8
VoxelNet	91.91%	89.98%	90.94%	5.0	95.67%	92.76%	94.20%	6.7
**Our Method**	**95.75%**	**98.22%**	**96.97%**	**35.9**	**98.11%**	**99.05%**	**98.58%**	**38.9**

**Table 4 sensors-23-08143-t004:** Comparison of the results of vehicle tracking algorithms using different datasets.

Datasets	Method	Evaluation Metrics				
		MOTA↑	MOTP↓	IDSW↓	ID-F1↑	ATSF↓
KITTI	Traditional Hungarian algorithm	63.15%	0.18	7	68.30%	0.009
	**Our method**	**88.12%**	**0.20**	**0**	**95.16%**	**0.018**
Matlab/ Simulink	Traditional Hungarian algorithm	69.38%	0.16	23	65.81%	0.007
	**Our method**	**90.56%**	**0.17**	**4**	**96.43%**	**0.021**

## Data Availability

The data in this paper is from KITTI’s object/3d object/velodyne pointcloud: https://www.cvlibs.net/datasets/kitti/eval_object.php?obj_benchmark=3d, downloaded from https://s3.eu-central-1.amazonaws.com/avg-kitti/data_object_velodyne.zip. The test scene is /testing/0001.

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
