# Peer review of "Vehicle Detection and Tracking with Roadside LiDAR Using Improved ResNet18 and the Hungarian Algorithm"

_sensors, 2023, doi:10.3390/s23198143_

Round 1

Reviewer 1 Report

The article proposes a vehicle detection and tracking framework based on roadside LiDAR. Firstly, a vehicle detection framework based on ResNet18 was developed, which can more effectively detect vehicles at full frame rate by improving the BEV mapping and optimizer's loss function. Then, the author proposes a new three-stage vehicle tracking algorithm that improves the Hungarian algorithm to better match objects detected in consecutive frames, while also proposing spatiotemporal logic and trajectory similarity to solve short-term occlusion problems.

Here are some suggestions:

1) The article explains three situations of vehicle matching during the tracking phase, and how to remove vehicles that disappear from view also needs to be explained.

2) The introduction section before each section in the article is too long and should be included in the related work section.

3) Suggest redesigning and beautifying the images and tables involved in the paper.

4) Given the closed source nature of MATLAB, it is recommended to re implement the simulation experiment in a Python environment.

Minor editing

Author Response

Dear Reviewers:

Manuscript ID: sensors-2589558

Title: “Vehicle Detection and Tracking with Roadside LiDAR Using Improved ResNet18 and Hungarian Algorithm

We wish to express our deep appreciation and appreciation for all of us, for your great efforts and suggestions for our manuscript. They are valuable and very helpful for revising and improving our paper and the important guidance to our research.

In this letter, we include a point-to-point response to your comments and the responses are underlined. The modification is marked in red in the revised version.

We tried our best to improve the manuscript and made some changes in the manuscript. These changes will not influence the content and framework of the paper. And here we list the changes We appreciate your warm work earnestly, and hope that the correction will meet with approval. Thank you for your time and patience. I am looking forward to receiving your letter.

Once again, we would like to thank you for your constructive comments and suggestions. Please feel free to contact us with any questions. We are looking forward to your reply.

Yours sincerely,

Authors

Reviewer 2 Report

General Overview

The paper discusses a study on vehicle detection and tracking by leveraging roadside LiDAR technology. It proposes a ResNet18-based vehicle detection architecture alongside a three-stage vehicle tracking algorithm. The study evaluates the proposed algorithms using both real-world (KITTI dataset) and synthetic (MATLAB/Simulink) datasets and argues for the superiority of their algorithms based on computational efficiency and effectiveness. The manuscript is well-structured, and the proposed algorithms show promising results. However, there are certain areas where improvement could make it a more robust contribution to the field.

Strengths

1. The manuscript is well-structured, making it easy to understand the complex algorithms discussed.

2. The study uses robust evaluation metrics, such as F1-Score and FPS, allowing for a comprehensive appraisal of the proposed algorithms.

3. The paper provides a comparative assessment with existing methods, which provides additional context for the efficacy of the proposed algorithms.

4. The study employs a multi-modal evaluation approach, utilizing both real-world and simulated datasets, which lends credibility to the results.

5. The manuscript acknowledges its limitations and suggests avenues for future work, enhancing the depth of the research.

6. The paper is topically relevant, considering the technological progress being made in the realm of autonomous vehicles.

Areas of Improvement

1. The abstract could be more concise, and key terms and abbreviations should be clarified for a broader audience.

2. The "Related Work" section could further elucidate the gaps in the current literature to emphasize the novelty of this study.

3. An in-depth explanation of performance metrics, such as F1 scores and Frames per Second (FPS), would be beneficial for contextualizing the results.

4. A granular comparison featuring specific benchmarks would add to the credibility of the paper's claim of outperforming existing methods.

5.  Minor grammatical errors should be corrected to improve the manuscript's overall readability (e.g., "it’s" should be "it is" for formal writing).

6.  The customization of the ResNet18 architecture for vehicle detection needs further elaboration for scientific completeness.

7.  Empirical evidence for handling occlusions should be included in the manuscript.

8.  A cost-benefit discussion regarding the use of 64-channel LiDAR would provide a more comprehensive view of the system's feasibility.

9.  Clarifications about the "external disturbances" in future work could set a more precise research direction.

10.  A discussion on the algorithm's adaptability to different sensor modalities would broaden the paper's scope and applicability.

Summary and Recommendation

The manuscript offers a notable contribution to the field of real-time vehicle detection and tracking using LiDAR technology. The study's structured layout, multi-dataset validation, and explicit acknowledgment of limitations make it a strong candidate for publication. However, the paper needs improvement in certain areas to enhance its scientific rigor and robustness. Therefore, the manuscript would benefit from addressing the identified areas for improvement, especially those related to methodological completeness, empirical substantiation, and resource feasibility analysis.

No comments here.

Author Response

(The authors gave the same response as above.)

Round 2

Reviewer 1 Report

The author has provided a comprehensive response to the comments and it can be accepted.

Minor editing of English language required